# Clinical Predictors of Preeclampsia in Pregnant Women with Chronic Kidney Disease

**DOI:** 10.3390/medicina56050213

**Published:** 2020-04-27

**Authors:** Bogdan Marian Sorohan, Andreea Andronesi, Gener Ismail, Roxana Jurubita, Bogdan Obrisca, Cătălin Baston, Mihai Harza

**Affiliations:** 1Department of General Medicine, Carol Davila University of Medicine and Pharmacy, 050474 Bucharest, Romania; bogdan.sorohan@yahoo.com (B.M.S); andreea.andronesi@yahoo.com (A.A); roxana_jurubita@yahoo.com (R.J.); obriscabogdan@yahoo.com (B.O.); drcbaston@gmail.com (C.B.); mihai.harza@gmail.com (M.H.); 2Nephrology Department, Fundeni Clinical Institute, 022328 Bucharest, Romania; 3Center of Uronephrology and Renal Transplantation, Fundeni Clinical Institute, 022328 Bucharest, Romania

**Keywords:** chronic kidney disease, preeclampsia, hypertension, proteinuria

## Abstract

*Background and Objectives*: Pregnant women with chronic kidney disease (CKD) are at high risk of adverse maternal and fetal outcomes. Preeclampsia (PE) superimposed on CKD is estimated to occur in 21%–79% of pregnancies. Both conditions share common features such as proteinuria and hypertension, making differential diagnosis difficult. Objective: The aim of this study was to evaluate the incidence and the clinical-biological predictors of preeclampsia in pregnant women with CKD. *Material and Methods*: We retrospectively analyzed 34 pregnant women with pre-existing CKD admitted to our department between 2008 and 2017. *Results*: Among the 34 patients, 19 (55.8%) developed PE and the mean time of occurrence was 31.26 ± 2.68 weeks of gestation. The median value of 24-h proteinuria at referral was 0.87 g/day (interquartile range 0.42–1.50) and 47.1% of patients had proteinuria of ≥1 g/day. Patients with PE tended to be more hypertensive, with a more decreased renal function at referral and had significantly higher proteinuria (1.30 vs. 0.63 g/day, *p* = 0.02). Cox multivariate analysis revealed that proteinuria ≥1 g/day at referral and pre-existing hypertension were independently associated with PE (adjusted hazard ratio = 4.10, 95% confidence interval: 1.52–11.02, *p* = 0.005, adjusted hazard ratio = 2.62, 95% confidence interval: 1.01–6.77, *p* = 0.04, respectively). The cumulative risk of PE was significantly higher in pregnant women with proteinuria ≥1 g/day at referral (log-rank, *p* = 0.003). Proteinuria ≥ 1 g/day at referral and pre-exiting hypertension predicted PE development with accuracies of 73.5% and 64.7%, respectively. *Conclusions*: Pregnant patients with pre-existing CKD are at high risk of developing preeclampsia, while proteinuria ≥ 1 g/day at referral and pre-existing hypertension were independent predictors of superimposed preeclampsia.

## 1. Introduction

Pregnant women with chronic kidney disease (CKD) are at high risk of adverse maternal and fetal outcomes, such as preeclampsia, low birth weight and death, even in cases of mild CKD [1,2]. Moreover, pregnancy itself is a risk factor for CKD progression [3]. Preeclampsia is both a systemic and hypertensive disorder in pregnancy. Estimated to complicate 2–8% of all pregnancies, it is a leading cause of maternal and perinatal morbidity and mortality [4]. According to previous studies, preeclampsia superimposed on chronic hypertension and CKD are estimated to occur in 26% and 21–79% of pregnancies, respectively [5,6]. Preeclampsia superimposed on CKD is challenging for the clinician because both conditions share common features such as proteinuria and hypertension, making differential diagnosis difficult [7]. There is some evidence regarding biomarkers associated with preeclampsia in pregnant women with CKD, but further studies are needed in this regard [8]. The aim of this study was to evaluate the incidence and clinical and biological predictors of preeclampsia in pregnant women with CKD.

## 2. Materials and Methods

### 2.1. Study Population

We conducted a retrospective cohort study which included all pregnant women with pre-existing CKD that were referred to our department, between 2008 and 2017. Exclusion criteria were: Absence of CKD diagnosis, absence of pregnancy, referral to our department after 20 weeks’ gestation and loss to follow-up. The final study population included 34 pregnant women with pre-existing CKD, out of 652 women of childbearing age monitored in our clinic between 2008–2017. All patients read and signed the informed consent form. The study was conducted in accordance with the Declaration of Helsinki, and the protocol was approved by the Ethics Committee of Fundeni Clinical Institute (No. 39622/ 12 December 2017).

### 2.2. Definitions

Preeclampsia was defined according to American College of Obstetricians and Gynecologists (ACOG) Task Force on Hypertension in Pregnancy (2013) and the revised statement (2014) of the International Society for the Study of Hypertension in Pregnancy (ISSHP) [9,10]. We considered the diagnosis of preeclampsia in CKD patients in every pregnant woman with pre-existing CKD who developed hypertension (≥140 mmHg systolic pressure or ≥90 mmHg diastolic pressure) after 20 weeks of gestation or a sudden exacerbation of preexisting hypertension or the need to improve antihypertensive therapy, plus the coexistence of one or more of the following new-onset conditions: proteinuria ≥ 300 mg/day or a significant sustained increase of preexisting proteinuria; liver involvement (elevated transaminases and/or severe right upper quadrant or epigastric pain); neurological complications (eclampsia, altered mental status, blindness, stroke, hyperreflexia when accompanied by clonus, severe headaches when accompanied by hyperreflexia, persistent visual scotomata); hematological complications (thrombocytopenia, disseminated intravascular coagulation, hemolysis); and fetal growth restriction. CKD was defined according to Kidney Disease Improving Global Outcomes (KDIGO) guidelines [11]. Referral was defined as the first visit of the patients to our clinic, between the beginning of pregnancy and week 20 of gestation.

### 2.3. Measurements

Clinical and biological parameters were collected from paper and electronic medical records. Renal function was evaluated based on serum creatinine and estimated with CKD-EPI formula. Proteinuria was measured from 24-h urine samples and reported in grams/day. Blood pressure was recorded by mercury sphygmomanometry and for the diagnosis of hypertension two determinations at least 4 h apart were taken into consideration.

### 2.4. Statistical Analysis

Values are reported as percentages for categorical data, mean ± SD for continuous parametric data, and median with interquartile range (IQR) for continuous nonparametric data. The chi-square, Student t-test, and Mann–Whitney U were used to evaluate the differences between groups. To analyze the clinical and biological determinants of superimposed preeclampsia, Cox regression analysis was performed, which included all the variables with *p* ≤ 0.10 from the group comparison. The final variables in the model were then selected using a stepwise backward elimination process. Kaplan–Meier curves were used to determine the cumulative risk of preeclampsia and the risk difference was analyzed via the log-rank test. To test the predictive ability of the variables, we calculated the sensitivity, specificity, positive predictive value, negative predictive value, positive likelihood ratio, negative likelihood ratio, and accuracy. A *p*-value of <0.05 was considered statistically significant. Statistical analysis was performed with SPSS version 20 (SPSS Inc., Chicago, IL, USA) and STATA version 14 (StataCorp, College Station, TX, USA).

## 3. Results

### 3.1. Characteristics of the Study Group

Among the 34 pregnant patients with CKD, 19 (55.8%) developed preeclampsia, 26 (76.5%) had CKD stages 3-4, and the mean time of occurrence was at 31.26 ± 2.68 weeks of gestation. The mean maternal age was 24 ± 3.96 years and 3 (8.8%) patients were over 30 years old. The frequency of nulliparity in our group was 85.3%. The mean value of creatinine at referral was 1.6 ± 0.6 mg/dL. The median 24-h proteinuria at referral was 0.87 g/day (IQR 0.42–1.50) and 47.1% of patients had proteinuria of ≥1 g/day. The most frequent primary disease for CKD was glomerulonephritis (76.5%). Eight patients (23.5%) had a body mass index (BMI) of ≥25 kg/m^2^ and half of the patients (50%) had pre-existing hypertension. None of our patients had a history of preeclampsia and there were no patients with CKD stage G5 (Table 1). Sixteen out of 19 (84.2%) patients developed PE < 34 weeks of gestation (early-onset PE) and in 3/19 (15.8%), PE occurred ≥ 34 weeks of gestation (late-onset PE).

### 3.2. Comparison between Women With and Without Preeclampsia

The characteristics of the women with superimposed preeclampsia on CKD and the differences between the two groups are summarized in Table 1. Patients with preeclampsia were more likely to be young (23.15 ± 2.31 vs. 25.66 ± 5.13, *p* = 0.06), hypertensive (63.2% vs. 33.3%, *p* = 0.08), dyslipidemic (31.6% vs. 20%, *p* = 0.44), and with decreased renal function at referral (46.16 ± 22.65 vs. 62.79 ± 21.4, *p* = 0.005) compared to those without preeclampsia. Furthermore, the preeclampsia group had significantly higher proteinuria (1.30 (IQR 0.67–1.50) vs. 0.63 (IQR 0.30–0.85) g/day, *p* = 0.02) and 13 of the 19 patients with preeclampsia had proteinuria ≥1 g/day at referral (68.7% vs. 20%, *p* = 0.005). Regarding fetal outcomes, fetal death occurred only in one pregnancy, the event taking place in the preeclampsia group (5.1%) and low birth weight was significantly higher in pregnant women with preeclampsia (84.2% vs. 6.7%, *p* < 0.001).

### 3.3. Predictors of Superimposed Preeclampsia

To assess variables associated with preeclampsia in CKD patients, Cox regression analysis with stepwise backward selection was performed (Table 2). Variables with *p* < 0.10 at the group comparison were included in the final model.

By univariate Cox analysis, proteinuria ≥ 1 g/day at referral was the only statistically significant variable as a risk factor for preeclampsia (*p* = 0.008), followed by pre-existing hypertension, which presented a tendency of significance (*p* = 0.07). In multivariate Cox analysis, proteinuria ≥1 g/day at referral and pre-existing hypertension after adjustment for age, nulliparity, and creatinine, were independent predictors of preeclampsia. The presence of proteinuria ≥1 g/day at referral and pre-existing hypertension increased the risk of preeclampsia 4.1 times (adjusted HR = 4.10, 95%CI: 1.52–11.02, *p* = 0.005) and 2.6 times (adjusted HR = 2.62, 95%CI: 1.01-6.77, *p* = 0.04), respectively.

In Figure 1A,B, the Kaplan–Meier curves show that the cumulative risk of preeclampsia at 35 and 38 weeks of gestation was significantly higher in pregnant women with proteinuria ≥1 g/day (68% and 81%, respectively) than in those with proteinuria <1 g/day at referral (27% and 34%, respectively) (log-rank, *p* = 0.003) and the cumulative risk of preeclampsia was also higher in pregnant women with pre-existing hypertension (67% and 70%, respectively) than is those without pre-existing hypertension (30% and 43%, respectively) (log-rank, *p* = 0.05).

We evaluated the predictive ability of the identified risk factors and found that proteinuria ≥1 g/day at referral and pre-existing hypertension could predict preeclampsia in pregnant women with CKD with an accuracy of 73.53% (95%CI: 55.6%–87.1%) and 64.71% (95%CI: 46.5%–80.3%), respectively (Table 3).

## 4. Discussion

Results from our study showed that pregnant women with CKD are at high risk of developing preeclampsia. Superimposed preeclampsia on CKD complicated 55.8% of the pregnancies in our study. This result is in agreement with other studies that reported values between 21% and 79% [6]. Moreover, the frequency of preeclampsia was increased in CKD stages 3–4. The increased incidence of preeclampsia compared to other studies could have been influenced by the diagnostic criteria we used, according to ACOG (2013) and the revised statement of ISSHP (2014), which exclude proteinuria as a mandatory condition for the diagnosis of preeclampsia [9,10]. The mechanisms by which CKD increases the risk of preeclampsia in not completely understood. Some mechanistical processes have been proposed, such as renin-angiotensin-aldosterone system (RAAS) dysregulation, endothelial dysfunction and complement dysregulation [12,13,14]. We reported no cases of postpartum preeclampsia within the first 6 months and no maternal deaths. However, one fetal death (5.3%) occurred in the preeclampsia group.

We evaluated possible clinical and biological predictors of superimposed preeclampsia in pregnant women with CKD and we found that the presence of proteinuria ≥1 g/day at referral and pre-existing hypertension increased the risk of superimposed preeclampsia 4.1 and 2.6 times, respectively. Other classic risk factors such as nulliparity, maternal age, BMI and the primary cause of CKD were not significantly associated with preeclampsia.

Proteinuria ≥1 g/day at referral was associated with a significantly increased cumulative risk of preeclampsia of 68% and 81% at 35 and 38 weeks respectively. Similar to our results, data from a prospective study, which included 504 pregnant women with CKD, showed that baseline proteinuria ≥1 g/day was an independent risk factor for pregnancy outcomes [15]. Also, data from a cohort study indicated that the proteinuria level at the beginning of pregnancy was strongly associated with subsequent preeclampsia in pregnant women with CKD [16]. A systemic review and meta-analysis which included 23 studies that evaluated the outcomes of pregnancy in CKD and CKD outcomes in pregnancy, demonstrated that proteinuria >0.5 g/day in pregnant women with CKD was a risk factor for superimposed preeclampsia [17]. Evaluation of proteinuria threshold at the time of referral in our study, until 20 weeks of gestation, is an expression of CKD and we proposed this clinical factor as a predictor and not as a diagnostic marker for superimposed preeclampsia.

According to our analysis, 63.2% of patients with preeclampsia had pre-existing hypertension. In agreement with our results, Braham et al reported that chronic hypertension increased the risk of superimposed preeclampsia 7.7 times in the US population and that the estimated incidence of disease was between 21% and 31.5% [5]. Another study by Webster et al. showed that the incidence of preeclampsia in women with chronic hypertension was 21% [18]. The higher percentage of preeclampsia in this subgroup could be explained by the fact that all these patients also had CKD and that a possible intrarenal renin-angiotensin-aldosterone system (RAAS) could be involved. It has been shown that in preeclamptic women, systemic RAAS components have lower circulating levels and there is an increased sensitivity to angiotensin II action, but little is known about the role of intrarenal RAAS [19,20]. Angiotensin II present in the renal tissue could be provided by angiotensinogen action, synthesized locally from tubular epithelial cells [21]. Furthermore, it has been demonstrated that angiotensin II promotes sFlt1 production in proximal tubular cells [22]. Yilmaz et al demonstrated that elevated intrarenal RAAS activity is involved in the development of preeclampsia, using elevated urinary angiotensinogen as a marker of local RAAS activation [23].

The diagnosis of superimposed preeclampsia on CKD is difficult to establish in clinical practice because of the common features shared by CKD and preeclampsia, thus making other parameters necessary for diagnosis. Promising results from previous studies showed that angiogenic (placenta growth factor (PIGF)) and antiangiogenic (soluble fms-like tyrosine kanise-1 (sFlt1)) factors and uteroplacental flows could be used as diagnostic tests for superimposed preeclampsia on CKD. However, they have not been implemented in clinical practice, making further study necessary [24,25,26]. More recently, plasma and urinary biomarkers derived from endothelial tissue (hyaluronan, intercellular adhesion molecule, vascular cell adhesion molecule, P-selectin, E-selectin), biomarkers of RASS and complement system activation have been studied in pregnant women with CKD. Kate Wiles et al. showed that increased plasma concentration of hyaluronan and vascular cell adhesion molecule had the potential to differentiate between pregnant women with superimposed preeclampsia on CKD and those without superimposed preeclampsia and that endothelial dysfunction had an important role in the pathophysiology of superimposed preeclampsia [13]. Relying on the latest evidence, the diagnostic utility of angiogenic, antiangiogenic and endothelial derived biomarkers in superimposed preeclampsia in CKD pregnant women is recognizable, but they are limited by availability and costs in clinical practice. Therefore, classical factors such as level of proteinuria at referral or pre-existing hypertension may still be useful in assessing the risk for superimposed preeclampsia on CKD [27] and could also contribute to the development of various prognostic tools alongside newly discovered biomarkers.

The strengths of our study are the use of adjusted models in the multivariate analysis and the diagnostic criteria used. However, its limitations such as the retrospective design, small size of the cohort study, and its single-center nature may limit its generalizability. Another limitation of our study that must be mentioned is regarding the selection bias. All patients included in the study had CKD and this could explain the higher incidence of PE than what was reported in other studies.

## 5. Conclusions

In conclusion, findings from our study indicate that pregnant women with pre-existing chronic kidney disease are at high risk of developing preeclampsia and that proteinuria ≥1 g/day at referral and pre-existing hypertension are independent predictors of superimposed preeclampsia. Consequently, maintaining proteinuria of <1 g/day in women with CKD who want to become pregnant could decrease the risk of preeclampsia. Moreover, women with pre-existing hypertension should be intensively monitored during pregnancy. However, to translate this conclusion into clinical practice is difficult considering the small size of our study population. Thus, larger studies are needed.

## Figures and Tables

**Figure 1 medicina-56-00213-f001:**
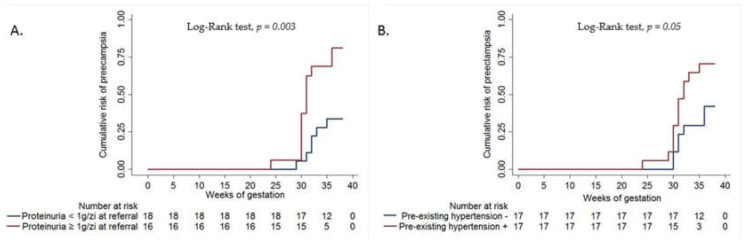
(**A**) Kaplan–Meier curves showing the cumulative risk of preeclampsia in pregnant women with proteinuria ≥1 g/day vs. <1 g/day at referral; (**B**) Kaplan–Meier curves showing the cumulative risk of preeclampsia in pregnant women with vs. without pre-existing hypertension.

**Table 1 medicina-56-00213-t001:** Characteristics of the study group.

Patients Characteristics	Overall (N = 34)	Preeclampsia (N = 19)	No Preeclampsia (N = 15)	*p*-value
**Maternal age (mean, years)**	24.26± 3.96	23.15±2.31	25.66±5.13	0.06
**≥30 years (%)**	3 (8.8%)	0 (%)	3 (20%)	0.02^a^
**Nulliparity (%)**	29 (85.3%)	18 (94.7%)	11 (73.3%)	0.07
**BMI (mean, kg/m^2^)**	21.91±3.36	22.28±3.26	21.45±3.54	0.48
**≥25 kg/m^2^ (%)**	8 (23.5%)	4 (21.1%)	4 (26.7%)	0.70
**Primary kidney disease (%)**	26 (76.5%)	16 (84.2%)	10 (66.7%)	
**Glomerulonephritis Tubulointerstitial disease (%)**	5 (14.7%)	2 (10.5%)	3 (20%)	0.20
**Diabetic kidney disease (%)**	3 (8.8%)	1 (5.3%)	2 (13.3%)	
**Pre-existing hypertension (%)**	17 (50%)	12 (63.2%)	5 (33.3%)	0.08
**Dyslipidemia (%)**	9 (26.5%)	6 (31.6%)	3 (20%)	0.44
**Hepatitis B virus infection (%)**	3 (8.8%)	2(10.5%)	1(6.7%)	0.69
**Hepatitis C virus infection (%)**	3 (8.8%)	1 (5.3%)	2 (13.3%)	0.41
**Urinary tract infections (%)**	17 (50%)	11 (57.9%)	6 (40%)	0.30
**Creatinine at referral (mean, mg/dL)**	1.6 ±0.6	1.79±0.75	1.36±0.40	0.05
**Creatinine at birth (mean, mg/dL)**	1.6±0.8	1.93±0.98	1.20±0.33	0.01 ^a^
**eGFR at referral (mean, ml/min)**	51.2± 22.8	46.16± 22.65	57.65±22	0.14
**eGFR at birth (mean, ml/min)**	51.5±21.8	42.62± 18	62.79± 21.4	0.005 ^a^
**CKD stage (%)**				0.23
**G1-G2**	8 (23.5%)	3 (15.8%)	5 (33.3%)	
**G3-G4**	26(76.5%)	16 (84.2%)	10 (66.7%)	
**Proteinuria at referral (median, g/24 h)**	0.87 (0.42−1.50)	1.30 (0.67−1.50)	0.63 (0.30−0.85)	0.02 ^a^
**Proteinuria ≥ 1 g/day at referral (%)**	16 (47.1%)	13 (68.7%)	3 (20%)	0.005 ^a^
**Fetal death (%)**	1 (3.1%)	1 (5.3%)	0 (0%)	0.27
**Low birth weight (%)**	17 (50%)	16 (84.2%)	1 (6.7%)	<0.001 ^a^

eGFR—estimated glomerular filtration rate; BMI—body mass index; CKD—chronic kidney disease; ^a^—statistical significance.

**Table 2 medicina-56-00213-t002:** Univariate and multivariate Cox regression analysis of predictors associated with superimposed preeclampsia on chronic kidney disease (CKD).

	Cox Univariate Regression			Cox Multivariate Regression		
Variables	HR	95% CI	*p*-value	HR	95% CI	*p*-value
**Maternal age**	0.88	0.75−1.03	0.12	-	-	-
**Nulliparity**	4	0.53−30.07	0.17	-	-	-
**Creatinine at referral**	1.57	0.87−2.73	0.13	-	-	-
**Proteinuria ≥ 1 g/day at referral**	3.91	3.74−9.98	0.008	4.10	1.52−11.02	0.005
**Pre-existing hypertension**	2.31	0.90−5.92	0.07	2.62	1.01−6.77	0.04

Cox multivariate regression with backward stepwise selection (variables introduced in the first step: maternal age, nulliparity, creatinine at referral, proteinuria ≥1 g/day at referral, pre-existing hypertension; variables that remained in the last step: proteinuria ≥1 g/day at referral, pre-existing hypertension); HR—hazard ratio; CI—confidence interval.

**Table 3 medicina-56-00213-t003:** Predictive analysis of risk factors for preeclampsia.

	Proteinuria ≥ 1 g/day at Referral		Pre-Existing Hypertension	
	Value	95% CI	Value	95% CI
**Sensitivity (%)**	68.42%	43.45%−87.42%	63.16%	38.36%−83.71%
**Specificity (%)**	80%	51.91%−95.67%	66.67%	38.38%−88.18%
**+LHR**	3.42	1.19%−9.85	1.89	0.86%−4.19
**-LHR**	0.39	0.19%−0.80	0.55	0.28%−1.10
**PPV (%)**	81.2%	60.09%−92.58%	70.59%	12.04%−84.15%
**NPV (%)**	66.67%	49.61%−80.25%	58.82%	41.77%−73.99%
**Accuracy (%)**	73.53%	55.64%−87.12%	64.71%	46.49%−80.25%

CI—confidence interval; LHR—likelihood ratio; PPV—positive predictive value; NPV—negative predictive value.

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
