# Peer review of "Clinical Predictors of Preeclampsia in Pregnant Women with Chronic Kidney Disease"

_medicina, 2020, doi:10.3390/medicina56050213_

Round 1

Reviewer 1 Report

Manuscript is well written.

Sample is small and this was well described in their limitations of the study.

Another limitation that I consider important to describe is the selection bias, as their population in the CKD clinic will have higher risk of presenting preeclampsia, and this is probably one the reasons that their incidence of preeclampsia is higher when compared to other studies.

Author Response

Dear Editor,

We would like to thank you for having revised our manuscript “Clinical predictors of preeclampsia in pregnant women with chronic kidney disease’’ (Manuscript ID: medicina-756748). We appreciate the constructive suggestions and comments from the reviewers.

In this revised version we tried to do our best to address all issues pointed-out by the reviewers, thus improving the quality of the manuscript. Specifically, we have introduced in the discussion section the limitation regarding selection bias as was suggested by the Reviewer no. 1. Also, we tried to respond to comments regarding low-dose Aspirin use, cases of early and late-onset PE and some fetal outcomes pointed-out by Reviewer no.2. As suggested by Reviewer no. 2 we have provided data for time of PE development and proportion of patients with early and late-onset preeclampsia. Moreover, we found missing % signs in Table 1 for Hepatitis B virus, Hepatitis C virus and Urinary tract infections and we completed accordingly.

We included a point-by-point response to reviewers’ comments and the changes will be highlighted in yellow in the main manuscript.

The revision has been made in consult with all coauthors and each author has given the approval for the final form of the manuscript.

Reviewer 1

We are thankful for your comments and recommendations. Regarding the limitation bias concern, we added this as a new limitation at the Discussion section, as you suggested. It is highlighted in yellow on Rows 218-220: “Another limitation of our study that must be mentioned is regarding the selection bias. All patients included in the study had CKD and this could explain the higher incidence of PE than what was reported in other studies.’’

Best regards,

Ismail Gener, MD, PhD, Associated Professor;

Fundeni Clinical Institute, Department of Nephrology;

Email address: gener732000@yahoo.com.

Reviewer 2 Report

The authors analyzed the maternal (and fetal) Outcome in 34 pregnancies with CKD and aimed to reveal predisposing factors for the development of preeclampsia.

According to their results, chronic Hypertension and proteinuria were indipendent risk factors of super-imposed preeclampsia´.

The manuscript itself is very well written, the statistical analysis well done and the results conclusive.

I have only minor comments:

As preexisting kidney diseases are a huge Risk factor for the development for preeclampsia, it is recommended to start with low doses Aspirin at first Trimester. Did the patients in your cohort recieve LDA ? If yes, starting when and how much (100 or 150 mg)? If no, why not ?

19 women developed preeclampsia: at which gestational age ? early or late onset ?

Were there any cases of IUGR ?

What About the neonatal Outcome ?

Author Response

Dear Editor,

We would like to thank you for having revised our manuscript “Clinical predictors of preeclampsia in pregnant women with chronic kidney disease’’ (Manuscript ID: medicina-756748). We appreciate the constructive suggestions and comments from the reviewers.

In this revised version we tried to do our best to address all issues pointed-out by the reviewers, thus improving the quality of the manuscript. Specifically, we have introduced in the discussion section the limitation regarding selection bias as was suggested by the Reviewer no. 1. Also, we tried to respond to comments regarding low-dose Aspirin use, cases of early and late-onset PE and some fetal outcomes pointed-out by Reviewer no.2. As suggested by Reviewer no. 2 we have provided data for time of PE development and proportion of patients with early and late-onset preeclampsia. Moreover, we found missing % signs in Table 1 for Hepatitis B virus, Hepatitis C virus and Urinary tract infections and we completed accordingly.

We included a point-by-point response to reviewers’ comments and the changes will be highlighted in yellow in the main manuscript.

The revision has been made in consult with all coauthors and each author has given the approval for the final form of the manuscript.

Reviewer 2

First of all, we appreciate your comments. Find our point-by-point answers below:

Low dose Aspirin. Current evidence based on meta-analysis and RCT recommended LDA (81-150mg/day) for prevention of PE in high risk patients. Our retrospective cohort included pregnant women with CKD and HTN which classified it as a high-risk cohort. Our general practice based on last evidences is to initiate Aspirin 100 mg/day in high-risk patients preferably until week 16 of gestation, but even if the patient was referred to our department after week 20, the use of Aspirin was nonetheless taken into consideration. Considering that we analyzed a retrospective cohort between 2008-2017, data regarding Aspirin use and dose were skewed and influenced by the guideline practice at that time. Moreover, not all patients provided in their records information regarding Aspirin use. For reasons related to missing data and different guideline practices, we cannot provide useful information regarding use of Aspirin (time and dose).

PE time of occurrence, early and late onset. The mean time of PE development was 31.26±2.68 weeks of gestation as we mentioned in the results section (Row 96). Sixteen out of 19 (84.2%) patients developed PE < 34 weeks of gestation (early-onset PE) and in 3/19 (15.8%), PE occurred ≥ 34 weeks of gestation (late-onset PE). We added this statement in the Results section (3.1 Characteristics of the Study Group) - Rows 103-105 (highlighted in yellow).

IUGR and neonatal outcomes. The aim of our study involved mainly preeclampsia and its associated risk factors. We do not have enough data in all patients regarding IUGR and neonatal outcomes in order to provide useful information.

Best regards,

Ismail Gener, MD, PhD, Associated Professor;

Fundeni Clinical Institute, Department of Nephrology;

Email address: gener732000@yahoo.com.